# “Compared to COVID, HIV Is Nothing”: Exploring How Onshore East Asian and Sub-Saharan African International Students in Sydney Navigate COVID-19 versus BBVs/STIs Risk Spectrum

**DOI:** 10.3390/ijerph19106264

**Published:** 2022-05-21

**Authors:** Sylvester Reuben Okeke

**Affiliations:** Centre for Social Research in Health, University of New South Wales Sydney, Sydney 2052, Australia; s.okeke@unsw.edu.au

**Keywords:** decision making, HIV, mental health, pandemic, public health response

## Abstract

*Background*: While a large body of evidence indicates changes in alcohol and other drug use among young people as a result of the COVID-19 pandemic, there is a lack of evidence around changes in sexual practices and how the pandemic may be impacting the potential spread of blood-borne viruses and sexually transmissible infections (BBVs/STIs). Most of what we know about sex during COVID-19 lockdowns is largely based on solitary sexual practices, which may not answer the critical question around how the pandemic may be shaping sexual practices among young people. Against this backdrop, this study explored how the COVID-19 pandemic may be shaping BBVs/STIs risk and protective practices among a sample of onshore African and Asian international students in Sydney, Australia. *Methods*: This phenomenological qualitative study involved semi-structured telephone and face-to-face interviews with 16 international university students in Sydney, between September 2020–March 2021. Generated data were coded using NVivo and analysis was guided by reflexive thematic analysis. *Results*: Participants reported elevated mental health distress because of the COVID-19 pandemic. Some participants reported engaging in casual sexual hook-ups as a strategy to mitigate the mental health distress they were experiencing. Some of these sexual hook-ups were condomless partly because COVID-related disruptions impacted condom accessibility. Additionally, the preventive practices of some participants who were sexually active during the lockdowns were focused on preventing COVID-19, while the risk of BBVs/STIs were downplayed. *Conclusions*: This study indicates a need for a comprehensive public health response to the evolving and near-endemic COVID-19 situation. Such a comprehensive approach should focus on empowering young people to prevent both SARS-CoV-2 and BBVs/STIs.

## 1. Introduction

The evolving Coronavirus Disease 2019 (COVID-19) pandemic continues to impact every aspect of human lives and activities. There are concerns that the pandemic may also adversely impact the progress that has been made so far in controlling the epidemics of blood-borne viruses (BBVs) such as HIV and sexually transmissible infections (STIs) [1,2,3,4]. Evidence suggests higher STI positivity rates, despite lower screening, in the COVID era compared to positivity rates before COVID-19 [3]. This indicates that research is needed to understand how the COVID-19 pandemic may be impacting sexual risk and protective practices, especially among young people. Exploring how the pandemic impacts on the risk of BBVs/STIs is important to inform comprehensive strategies during a public health crisis, which focus not just on the health problem, but also on possible spill overs. For instance, the COVID-19 pandemic may result in elevated psychological distress, which could, in turn, trigger unsafe sexual practices and alcohol use. This is especially important for vulnerable populations such as migrant groups, including international students.

Pre-pandemic evidence has shown that international students in developed countries such as the UK [5], US [6] and Australia [7] experience social isolation, loneliness and other psychological stressors that impact on their mental health and well-being. Australian evidence indicates that social isolation has the strongest negative effect of all COVID-related factors on university students’ psychological well-being [8]. Compared to the general population, international students are more likely to experience fewer family interactions [9] and a climate of uncertainties and xenophobic reactions, which could aggravate the experience of COVID-related mental health distress [10].

With or without a pandemic, people may generally seek relief from mental health distress by engaging in pleasurable activities such as alcohol and other drug use [8,11,12,13,14,15] and sex [16,17]. Pre-COVID, social isolation and migration-related distress have been linked with risky sexual practices and poor sexual health outcomes among international students in Australia [18,19]. Given that social isolation and distress are known risk factors for practices that risk BBVs/STIs, it is important to gain more understanding of international students’ experience of psychological distress during the COVID-19 lockdown, whether sex was used to seek relief and how the COVID versus BBVs/STIs risk spectrum was navigated.

While a large body of evidence indicates increased alcohol and other drug use among young people because of the psychological distress associated with the COVID-19 pandemic, especially during and after the restrictive lockdowns [8,12,13,14,15], evidence around the use of sex to mitigate the psychological impact of the pandemic is scarce. Most of what we know about sex during COVID-19 lockdowns is largely based on solitary sexual practices [16,17,20], which may not answer the critical question around how the pandemic may be shaping BBVs/STIs risk practices among young people. Therefore, understanding how international students navigated the COVID-19 versus BBVs/STIs risk spectrum during COVID-19 lockdowns could help improve what we know about the impact of the pandemic on BBVs/STIs risk practices among young people. This insight could inform how this population can be supported to act to protect themselves against both severe acute respiratory syndrome coronavirus 2 (SARS-CoV-2) and BBVs/STIs. This is necessary as COVID-related anxiety may remain after lockdown and sex may continue to be one of its mitigation strategies.

Therefore, beyond exploring experiences of psychological distress due to COVID-19 lockdowns among international students [8,10], the present study also investigates whether and how their sexual practices, in relation to the risk of and protection from BBVs/STIs, are being shaped by the pandemic. Improving sexual health knowledge among young people remains a central focus of many interventions. While this is important, it is also beneficial to recognise that knowledge is not enough to trigger behaviour change. Therefore, beyond knowledge, it is important to understand whether and how emotions, intuitions and experiences shape BBVs/STIs risk and protective practices during the COVID-19 pandemic, as we still do not fully understand how these affective states shape health behaviours [21].

Thus, the present study, also aims to improve the understanding of the pattern of decision-making processes that shape risk stratification and preventive practices. Decision-making patterns that shape young people’s preventive sexual practices could be explored from two theoretical standpoints—affective and rational decision-making models [22,23]. An affective decision-making process entails making decisions while relying on intuition, emotion and experiences. This decision-making model uses heuristic strategies, i.e., “rule of thumb” to compensate for the lack of adequate information, knowledge or time to think through all options [23]. A typical example of such a thinking pattern is demonstrated when young people fail to use a condom in a penetrative sexual encounter because they are having sex with a “known”, “trusted” or healthy-looking partner [23]. Conversely, the rational decision-making model is foregrounded on the possession of adequate knowledge and relying on reasoning, negotiation and deliberations in deciding health practices or behaviours [22].

Improved understanding of the decision-making process that shapes practices could inform priority areas for further research, policy and interventions. This way, we may be more able to tackle the current COVID-19 pandemic while not losing focus on the United Nation’s General Assembly [24] and the WHO [25] target of eradicating viral hepatitis, HIV and STIs as public health threats by 2030.

## 2. Methods

### 2.1. Paradigm and Design

This qualitative study is part of a larger mixed-methods research that investigated BBVs/STIs risk and protective practices among East Asian and sub-Saharan African international students in Sydney, before and during the COVID-19 era. Theoretically, the study is foregrounded on an interpretivist paradigm and methodologically guided by interpretative phenomenological analysis (IPA) [26]. A more detailed description of the research design and methods have been published elsewhere [27].

### 2.2. Sample and Recruitment

Sixteen onshore East Asian (EAS) and sub-Saharan African (SSA) international students who were in Sydney during the first COVID-19 lockdown (March–June 2020) provided data for this study. Data were generated through a combination of telephone (fourteen) and face-to-face (two) semi-structured interviews conducted between September 2020–March 2021. The data collection period was extended because of pandemic-related difficulties in reaching and recruiting participants. Participants were eligible to participate if they were studying on a student visa and came from an East Asian or a sub-Saharan African country. Participants were also deemed eligible if they were aged 18 years or above, were in Sydney during the first COVID-19 lockdown and had spent at least three months in Australia. It is instructive to note that some data collection began in September 2020 when there was no COVID-19 vaccine.

Convenience sampling was used for recruitment as participants were recruited using electronic flyers. These flyers were used to advertise the study on various international student groups’ social media pages. To avoid coercion, participants were not approached directly to participate in the study. Rather, the study was actively advertised across different international student groups’ online pages and information on how to reach the researcher provided, in case any eligible student wished to participate. International students who volunteered to participate contacted the researcher via the email address or phone number provided to arrange interviews. While arranging interviews, selecting a time and/or setting where there would be privacy to discuss sensitive issues, such as one’s sexual practices, was emphasised. All the 14 participants interviewed over the telephone participated from their homes. The interviewer also conducted the interviews at home using the “home office”—a spare room that now serves as an office following the work from home arrangement. Interviews conducted face-to-face were conducted at an open setting on two separate university campuses. The settings were carefully chosen to ensure privacy and adequate space to fully observe COVID-19 safety protocols.

The University of New South Wales Human Research Ethics Committee (more than low risk) granted ethical approval (HC190215) for this study and the study was conducted in strict adherence with all the relevant regulations and guidelines approved by this Institutional Review Board. All the participants provided verbal or written informed consent, depending on the nature of the interviews. All the 14 participants interviewed over the telephone provided verbal consent, while the two participants interviewed face-to-face provided written consent. Personal or identifying information was not collected during the interviews. Informed consent also involved participants understanding their right to voluntarily participate and to withdraw at any stage of the study and that de-identified results of the study would be published in academic journals. Considering the impact of the COVID-19 pandemic on mental health and that some participants may have experienced distressing situations that could affect them psychologically before or after the study, participants were provided with contact details of free professional psychologists and counselling services. Participants who mentioned their country of origin during the interviews had these country names de-identified. To further ensure strict confidentiality, participants were described in the study administration document using a coding system that included background (whether East Asia or sub-Saharan Africa), the gender they identify as, age and the sequence of participation. As such, a 23-year-old participant from East Asia, who identifies as a female and was the second in the sequence of interview, was coded as: EAS/f/23/2.

### 2.3. Data Collection

In line with the participant-oriented approach of interpretative phenomenological analysis, data collection involved participants telling their own stories, from their own perspectives and in their own words [28,29]. This was facilitated using semi-structured interviews (attached as Appendix A). This method of data collection was useful in ensuring flexibility to accommodate and encourage participants to speak about other contexts and explore issues they felt were also important in the context of the study [28]. Participants were asked to share their lived experiences, their observations of the experiences of other international students from their backgrounds and those in their social networks, in relation to mental health and sexual practice during the COVID-19 restrictive lockdown. Participants received an AUD 20 gift voucher as reimbursement for costs that may have been associated with their participation.

### 2.4. Data Analysis

Interviews were audio-recorded, transcribed and coded into NVivo for analysis. Participants’ experiences were thematically analysed [26,30] using reflexive thematic analysis [31]. Analysis involved data immersion and familiarisation through multiple and iterative reading of the transcripts while listening to the audio-recorded interviews. The procedure for coding combined deductive and inductive strategies [30,31]. The affective–rationale model of decision making [22] was used in making sense of the interview data and framing some perspectives and practices shared by the participants. Initial themes were generated from the codes and then reviewed and refined to create the final themes. Relevant quotes from the data are reported to support the themes.

## 3. Results

### 3.1. Socio-Demographic Characteristics of Participants

Half of the participants (8 of 16) were aged between 19–24 years, six were aged 25–29 years while two were aged 30 years and above. Most participants (15 of 16) were never married and identified as female (9 of 16). Of the 16 participants, nine were from a sub-Saharan African background while seven were from an East Asian background (Table 1).

### 3.2. Generated Themes

Patterns in the generated data showed three overarching themes: (a) elevated mental health distress, (b) sex as a strategy for relieving mental health distress and (c) navigating the COVID-19 versus BBVs/STIs risk spectrum.

#### 3.2.1. Elevated Mental Health Distress

Almost all the participants reported increased mental health distress because of the COVID-19 pandemic. The elevated level of emotional distress reported by participants could be attributed to two intersecting factors—(a) social isolation caused by lockdown and (b) COVID-related anxiety.

(a)Social Isolation Caused by Lockdown

Based on the accounts of many participants, mental health distress among international students was partly caused by the social isolation. This isolation was as a result of the breakdown in social interactions and human contacts due to the restrictive lockdown that became a strategy for containing the spread of COVID-19 [14,15]. While the feeling of lockdown-related social isolation is not peculiar to international students [8], this population may have had higher vulnerability considering that they were far away from home at a time of a profound global uncertainty:


*Because, last year was my second year in Australia. So, I was just starting to have few friends and colleagues and trying to have the life of an international students—having a few friends. So, at a time when I started feeling a little bit normal, as if I was back home with friends and people to mingle with, we lost contact due to COVID. So, it was quite hard for me. It was a big shock because it is not easy not being able to communicate with anyone. Everything turned into online communication…So, this affected me and the way I see things. I was quite depressed*
(female/26/EAS)

Although there were options for virtual communications and interactions, participants generally reported preferring human contacts to virtual interactions. Moreover, considering that the transition to virtual interactions happened so quickly, there was little time for people to adjust from face-to-face interactions and classes to the online mode. This quick transition affected participants:


*…everyone was in lockdown, the only way you can communicate was via this type, I guess phone, making a call or whatever and…Like with some people, like for me personally, I prefer meeting people in-person to talking over the phone. So, during lockdown, it affected me mentally because it is a shift*
(female/21/SSA)

International students may not have family members in Australia and this could have increased their vulnerabilities to mental health stress. It is doubtful whether virtual communications could effectively mitigate the level of psychological distress that international students were facing at the outset of the pandemic when there was no vaccine or adequate scientific knowledge about the novel virus. Additionally, international students from lower- and middle-income countries, which were the settings from where most participants were drawn, may not have had reliable and affordable internet connections back home compared to international students from developed countries. Vulnerability around social isolation was therefore prominent in participants’ reported experience of mental health distress.

Another area of vulnerability that was cited during the interviews was the feeling of xenophobic reactions among some international students from settings where COVID-19 was first detected. This may also contribute to mental health distress as students who felt they were victims of xenophobia may have experienced higher mental health distress:


*A lot of [international students]…especially us, Asian students, we felt stigmatized because many people didn’t like that the disease came from where we come from. So, a lot of us thought of just going back home to find a better environment and [a] more supportive one*
(female/26/EAS)

(b)COVID-19 Anxiety

As well as social isolation, high mental distress could also be attributed to anxieties, concerns and uncertainties about the COVID-19 pandemic. There were and still are, concerns around the effect of the pandemic on national and global economies, the job market and career goals. This anxiety also impacted, and still impacts, participants’ motivation to study and plan for future endeavours:


*there is this point that I felt really stressed out…I am almost graduating…looking that “I have no future at the moment and I am just studying this hard, it is not worth it”. I did have that thought and it is still growing…although the lockdown is no longer there but these worries, these stress[ors] are still there…a lot of my friends are also having this concern as well. One of them is really depressed to the point that they actually looked for a psychologist. It is pretty bad*
(female/20/EAS)

There were also concerns about what family members back home were experiencing and how the pandemic was impacting them. This was especially the case considering that participants were recruited from lower- and middle-income countries, where government support to cushion the socio-economic impact of the pandemic may not have been as robust as in more developed economies:


*I was more concerned about…people back home, how they cope and everything*
(female/30/SSA)

The evolving nature of the pandemic was also cited as a major stressor, considering that there seemed to be no end in sight. International students with time-bound visas have strategic plans and career goals that are more vulnerable and may therefore feel more concerned and anxious as the pandemic lingers on. The mental health distress occasioned by this concern impacted sleep and academic productivity:


*…and the biggest problem was just hearing that there was no end into this, no one knows how long it will last. The end was unknown, and it was completely devastating for me …I feel so much tired and not as productive I should be*
(female/26/EAS)

Participants reported that there were no adequate supports or tailored mental health services for international students. Though some levels of support were cited, there was the common view that these supports were not adequate to address the enormity of mental health distress that international students were experiencing. The experience of one participant indicates that available services were general in nature and not tailored to the specific needs of international students from conservative settings, where mental health issues may be stigmatized [7]. This may have affected the confidence and extent to which some of these students would seek mental health care, engage and openly communicate their mental health concerns:


*Honestly, we [international students] were lost, and I think we are still. Yes, we got supportive emails from time to time, and they direct you to mental and health clinics, but there is not much that can be done there. I had spoken to someone online, just over the phone, because I was too embarrassed to go and say, “I am just stressed and scared because I don’t know how things are going to be.” The phone call was just to calm me down and give me tips on how to get a better sleep. It was just a talk. So, I didn’t call again*
(female/23/EAS)

#### 3.2.2. Sex as a Strategy for Mitigating COVID-Related Mental Health Distress

Sex was one of the commonly cited strategies for getting relief from the psychological distress associated with the COVID-19 pandemic. Perspectives shared by some participants suggest that even without a pandemic, sex may be perceived and used as a “treatment” for stress. This makes the pandemic “double stress” for such people, because COVID-related restrictions may impact access to regular and casual sexual partners:


*…for me, sex is one of the best treatment[s] for stress. People get frustrated for so many things but when sex is involved, it calms the situation down*
(male/29/SSA)


*There is this one friend, specifically, she is kind of, not addiction, but when she has a problem or she is stressed out, she usually use that [sex] as a way of relieving her stress and now it is like kind of double stress, you have the pandemic, and you can’t have sex*
(female/20/EAS)

The heightened stress level that international students experienced may have reinforced their desire to use sex to seek relief from anxiety; especially at the outset of the pandemic, when there was a considerably higher level of uncertainty:


*that kind of depression and anxiety will influence your sexual desire and your sexual behaviour. So, I do believe that most [international students] will [use sex as a relief]. I think so because when you get a successful sex, then you will be happy, [a] sense of happiness will come to your mind … when you have sex, you don’t think of anything else*
(female/24/EAS)

One participant linked sexual attraction at that time to a feeling of worth, which could boost self-image and mental health:


*Getting involved with someone intimately in the world that is full of uncertainties, can give you the feeling that you are very important or significant to certain people. It might be giving us more certainty and create more mental stability… In an era of uncertainty, that sense of belonging which comes through sexual involvement is a way to tell us that we are still very important or very significant in certain people’s life*
(male/31/SSA)

Some participants also shared their own personal experiences of using sex as a strategy to mitigate the mental health impact of the COVID-19 pandemic.


*I [started] using sex to clear my head… clearing off my mind and trying to get my mind off from everything that is going on*
(male/19/SSA)

There appears to be a feeling of shared mental health distress and shared need for sexual interactions among participants who reported casual hook-ups during the lockdown. Perspectives during the interviews indicate that this sense of mutual need for sexual interaction may impact decisions to use condoms in such hook-ups. This is because some participants may perceive that the person, they are having a hook-up with is only seeking casual sex because the pandemic stress is making them to do so. The potential partner may therefore be perceived as not having multiple or casual hook-ups before the pandemic and, therefore, less likely to transmit BBVs/STIs to them.


*I was meeting people at the time… we met in either their places or mine for sex…those people I am meeting are going pretty much through the same thing, and I think they are looking forward to a chance to have sexual connections…so, it is a mutual interest*
(female/23/EAS)

Though the popular view was that sex could be an important strategy for mitigating the impact of COVID-related mental health distress, some participants expressed contrasting views. Specifically, one participant with an opposing view mentioned that sex-based relief is only momentary as the mental health issue remains.


*Being sexually involved with someone does not take away the problem or challenge. After that whole euphoria, the challenge is still there, you still go back to that point*
(male/24/SSA)

Similarly, there is also the view that people who are facing mental health distress would not even think of sex as COVID-19-related anxiety and depression may affect sexual drive.


*When someone is depressed, sex is not important thing because for me, if I am depressed, I usually don’t want to have that [sex]…although it [sex] is relaxing, but I think sex should happen when people are relaxed but if someone is under a very high pressure, they won’t be able to have sex*
(male/27/EAS)

While using sex to seek relief from psychological distress is not unusual, there are potential concerns around risk practices that may be specific to the COVID-19 pandemic. For instance, there could be concerns with recruiting sexual partners since there was a restrictive lockdown. Perspectives shared during the interview indicate that this lockdown could impact access to regular sexual partners, thereby increasing the chances for casual hook-ups that may be condomless. In addition, the interview data showed that decisions for safer sex may be impacted by impulse and poor judgment because of the psychologically distressing experiences that some international students may be facing. This could increase the likelihood of ignoring sex-related risks for the sexual gratification of having relief from the pandemic-related mental health distress:


*…with the whole stress of COVID–19, people were feeling like giving up, you don’t know what will happen next. Everybody is looking for someone to get consolation, everybody is really stressed. People could jump on each other without thinking…and they could have spontaneous reaction…because when you have so much stress level you might not even be thinking straight*
(female/30/SSA)

The interviews also showed that disruptions to the distribution of essential commodities also affected condom accessibility at that time. This disruption may increase condomless sex as pre-COVID evidence shows that some members of this study population engage in condomless sex because of some complexities around condom availability and accessibility [32].


*There is a huge restriction in terms of how people even go to buy things in the stores. It is not like before that I can just be coming back from class, from the uni [university], I can just get a condom from the store or walk [in]to a clinic and get condoms*
(male/31/SSA)

Though one participant mentioned proactiveness by acquiring some condoms before the disruptions, there may still be concerns with adequacy in supplies and with restocking. At that time, essential commodities were off most shelves because of panic buying and COVID-related disruptions to supply chains. Interestingly, condom availability may have also been impacted by its use as a hand glove for protection against SARS-CoV-2.


*Before COVID-19, I bought a packet which is like 12 or 14 [pieces of condoms], and during the period when everything was off the shops, I was not affected…condom in many supermarkets are off the shelves because many people just buy it and use it as a glove…they use it in the finger as they need to touch some screens or buttons in public place. I think some people resorted to buying condoms for protecting their hands and this affected the availability of condoms*
(female/24/EAS)

Therefore, condom inaccessibility may have contributed to reinforcing condomless sexual practice with casual partners during the lockdown as reported by one participant who had such an experience:


*I met around three people and we met a few times. {Probe: Ok, and they were all protected? I mean with condoms?} Not exactly. Not every time. It was just the lack of…there was none, he couldn’t find any…he couldn’t buy any*
(female/23/EAS)

As well as inaccessibility, the perception that consistent condom use is unrealistic may also reinforce condomless sex in hook-ups during the lockdown. Moreover, heuristics—such as a casual partner’s self-report of sexual practices and assessing BBVs/STIs status by physical appearance [23]—seem to impact condom use. Similarly, the heuristic of developing trust with time also shaped decisions to practice condomless sex:


*…[the] people I met more often, I know they don’t have any other sexual partners, based on what they told me. It is just the pandemic that is causing them to [be] craving sexual intercourse because their partners were locked down somewhere else. The feeling gave me a lot of security. Those people are not too thin, they don’t have any usual symptoms, their body looks healthy to me. So, I was thinking the percentage of them transferring any disease to me will be quite low. So, I felt quite safe…and it wasn’t just that, the idea is, I don’t think we needed a condom every time we had sex. The first time, obviously yes, we used a condom, but by the second and third time, I said it can be ok to do it without [condom]”*
(female/26/EAS)

Moreover, another potential concern around seeking mental health relief through sex is that participants may be mixing sex with alcohol:


*…if we are not going outside for party or [to] pubs, people want support. So, your neighbours can just organise a drink. So, after alcohol, sex happens*
(male/23/EAS)

Mixing alcohol and sex in such circumstances may also weaken condom use decisions if there is intoxication. This may also increase the risk of the sexual transmission of blood-borne viruses and other infections.

#### 3.2.3. Navigating the COVID-19 versus BBVs/STIs Risk Spectrum

One cardinal aim of this study was to understand the decision-making pattern that shaped how participants ideate and navigate the COVID-19 versus BBVs/STIs risk spectrum. The interview data showed a pattern of risk stratification in participants’ perceived vulnerability and the severity of COVID-19 in comparison with BBVs/STIs, which shaped their overall preventive practices. The combined role of the rational–affective decision-making process was apparent in participants’ perceptions and preventive practices.

Perspectives shared by this study’s participants indicate that risk conceptualisation and protective behaviours against both SARS-CoV-2 and BBVs/STIs were affect-driven (emotion, intuition, experience) or rational-driven (reasoning, negotiation, deliberation) or both. For instance, both of these thought processes shaped one of the participants’ reported decision to cut down on the number of partners during the lockdown for fear of contracting SARS-CoV-2. Meanwhile, when making partner choices, affect or heuristics became evident. For instance, relying on observable symptoms to decide on the choice of a sexual partner or whether to have sexual interaction with a potential hook-up partner demonstrates affect-driven decision making. Since both SARS-CoV-2 and BBVs/STIs can be transmitted by an asymptomatic person, such decision making may not be effective:


*…you are meeting less people, so, you don’t get to meet someone new or even someone you use to know because you are never too sure of who has the virus (SARS-CoV-2). So, it’s better to stay safe. {Probe: This period, did you ever think of STIs, BBVs, I mean HIV, was it a concern to you at any time or was it just COVID?} Basically, it was COVID. When we met, I was checking if they had symptoms, [if] no[t], it was ok [to have sex]*
(female/23/EAS)

Similarly, the affective decision-making process around acting in line with situational expectations (preventing SARS-CoV-2) could also have shaped participants’ decisions and preventive practices at that time. Consequently, preventive efforts during the lockdown were more focused on ruling out chances of contracting SARS-CoV-2, while largely ignoring the risks of BBVs/STIs, for most of the participants who were sexually active at that time:


*the whole noise around the world is on COVID-19, that’s what everyone talks about. So, if I know that I am not going to get COVID, I might forget about prioritizing other things that were bothering issues in the past and try to focus on [preventing] COVID*
(male/31/SSA)

Moreover, in line with affective decision-making patterns, there was also a widely held view that while BBVs/STIs are manageable, COVID-19 might not be. It is noteworthy that data collection began when there was no vaccine for COVID-19. Thus, participants’ views and practices in this regard, may have been shaped by experience, intuition and emotion around the rule of thumb that COVID-19 is more severe when compared to BBVs and STIs. This perception of COVID-19 as a “death sentence” may make BBVs/STIs less of a concern and something that can be ignored once the risk of contracting COVID-19 has been ruled out.


*COVID is a big disease when compared with HIV because COVID kills immediately. COVID is a big disease, the world has never seen anything like this…when compared to COVID, HIV is nothing…if the person [sexual partner] is COVID free, you can do sex with or without a condom, it doesn’t matter*
(male/23/EAS)

This viewpoint is also succinctly captured by another participant who stressed that BBVs/STIs are less of a concern because they are treatable or manageable:


*The main centre of everyone now is COVID-19, because for STI, you still know that there is management, you still have a treatment, but for COVID, there is no treatment. So, you know once you have it, you are gone. If you have sexual disease, you can still treat it or manage it but COVID, it’s a no no*
(female/28/SSA)

One affect-driven dominant pattern of reasoning among participants that is of concern in relation to the risk of BBVs/STIs, is the heuristic view that having sex with a neighbour or someone within the same vicinity is safe. Though such arrangements may minimise the risk of contracting SARS-CoV-2, risks around STIs and the sexual transmission of BBVs, may still be present:


*given that there is a huge restriction, if you have someone that is within your unit or within your environment…the fear or the risk of being exposed to the virus [SARS-CoV-2] is greatly reduced because you are within [the]same circle, same vicinity with these people. It will be a good opportunity to mingle with those people…getting involved with them sexually might be a safer option than running into the city*
(male/31/SSA)

The concern about ignoring the transmission of BBVs/STIs when arranging to have sex with people one can have access to because of the lockdown, may be further compounded by the perspective that BBVs/STIs are more likely to be transmitted by strangers. This view that BBVs/STIs are less likely to be transmitted by people we know and in home settings—compared to in bars and pubs—could further interfere with the intention to use a condom since the chances of transmitting BBVs/STIs by people in the neighbourhood is perceived to be low:


*people are not using condoms because people don’t find people from outside. They are always inside because of the restriction, so, they won’t meet a new person, like going somewhere and find a new person. They are staying in their home, having sex with their neighbour…*
(male/23/EAS)


*if you are having sex with people around, they are not strangers or people you don’t know…when you have sex with strangers, people you don’t know, the possibility to spread disease [BBVs/STIs] can be higher*
(female/28/SSA)

Notably, one of the study participants decried the absence of any conscious measures to ensure that people who are having casual hook-ups during the lockdown are sensitized and supported to practice such hook-ups in a safe and protective manner:


*during the whole COVID-19 lockdown, there is COVID in the air, every other disease just left just like that. Nobody was talking about how to transmit HIV, how to transmit STIs. Nobody was talking about it in the sense that people have sexual relations in their houses. You have communal [living arrangement] and nobody was talking about how to prevent people from contracting HIV and STIs. There were no measures put in place to ensure that people are safe in terms of BBV and STI*
(female/26/SSA)

The absence of such a public health strategy may further increase the possibility of ignoring the risk of BBVs/STIs while seeking sexual partners and engaging in casual sexual relationships during restrictive lockdowns. This may potentially increase the risks of BBVs/STIs.

## 4. Discussion

The overarching aim of this study is to understand whether and how the evolving COVID-19 pandemic is shaping BBV/STI risk and protective practices among young people, with a focus on international university students in Sydney. The results of the study showed that international students from East Asian and sub-Saharan African backgrounds experienced elevated level of psychological distress, and since this level of distress may have co-existed with inadequate and less-contextualised mental health services and support, they may have sought relief using pleasurable strategies, such as sex. Moreover, preventing SARS-CoV-2, with lesser regard to BBVs/STIs, was reported to be the focus of preventive practices.

The result of this study showed that mental health distress associated with social isolation and COVID anxiety was common among the study participants. Though pandemic-related distress is not restricted to international students, as it is also commonly reported among the general population [33,34,35,36] and young people [37,38,39,40,41], the level of distress faced by international students may be higher because of intersecting vulnerabilities including the socio-economic impact of the pandemic on daily sustenance and xenophobic reactions [10] as well as disruptions to learning [42,43]. Considering that the study population are young people from largely conservative cultures where mental health issues are stigmatized [7], not having access to adequate or contextualized mental health services and support could increase the likelihood of seeking relief from pleasurable activities such as sex, as reported by some participants. Thus, the use of sex as a strategy to seek relief from the mental distress associated with the pandemic was reported by some participants.

Using sex as a strategy for relief from pandemic-related mental health distress poses concern for the risk of BBVs/STIs as found in the present study and supported by previous ones. First, young people, generally, do not use condoms consistently [44,45,46] and this also applies to this study population [32]. Therefore, the pre-pandemic practice of inconsistent condom use may increase the risk of transmitting and contracting BBVs/STIs. Second, and importantly, pandemic-related disruptions to supply chains and health services impacted access to condoms at shopping malls and pharmacies as well as at hospital settings. These disruptions impacted condom accessibility and contributed to condomless sex. Pre-pandemic, condom accessibility was reported as one of the issues contributing to condomless sex among the study population [32]. In the face of the disruptions to supply chains and panic buying associated with the pandemic, the pre-pandemic condom accessibility issue may have become compounded, thereby increasing the likelihood of condomless sex during the pandemic and the risks of the spread of infections. Third, stressors [47], including COVID-19 pandemic-related stress [48], affect decision making, resulting in poor and impulsive decisions [49,50] that may impact condom use. Moreover, the possibility of mixing sex with alcohol while seeking relief from mental health distress, as indicated during the interviews, could also impair decisions to make safer sex-related choices as alcohol may impair the condom use decision [32].

Sexual activities in a pandemic between people who are not regular and/or cohabiting partners could involve the risk of transmission of SARS-CoV-2 and BBVs/STIs. The present study unpacked how decisions around preventive practices for these two risks are shaped. The study’s result indicates that navigating the COVID-19 versus BBV/STI risk spectrum was largely shaped by heuristic views, culminating in paying lesser attention to preventing BBVs/STIs. The perceptions and self-reports of participants who were involved in casual sexual hook-ups indicate that COVID-19 was perceived as more severe than BBVs/STIs. As such, preventive efforts were largely focused on ruling out the possibilities of contracting SARS-CoV-2 as compared to BBVs/STIs, which are known to be treatable or manageable. This pattern of decision making exemplifies the affective model in which experience and intuition inform decisions and health practices [22]. Considering that COVID-19 was novel, and scientific knowledge about the virus was still only just emerging at that time, young people may have had the tendency to prioritise preventing it over BBVs/STIs, which they knew could be cured or managed, as opposed to incurable.

## 5. Limitations and Implications

Data collection began at a time when COVID-19 vaccines were not available, therefore, it is important to note that the level of mental health distress reported at that time may be different from what it was after the vaccine roll out. Again, participants volunteered to participate in this study, and, as such, the experiences shared may not reflect the experiences of the international student population from which the sample was drawn.

The result of this study has implications for designing, implementing and evaluating effective interventions that not only address sexual health knowledge and attitudes, but also the complex and affect-based perspectives that shape young people’s BBV/STI risk and protective practices whether in a pandemic or not. The pre-pandemic sexual practices of young people may have been negatively impacted in the pandemic where young people maintained sexual connections that may also have involved risk taking [51]. The view that COVID-19 is novel and should be prioritised over BBVs/STIs may serve the purpose of using the pandemic to rationalise usual pre-pandemic low BBV/STI protective practices. As found in the present study and supported by previous ones, engaging in condomless sex because of the perceived safety with a partner [52,53], condom-related fatigue or trust [23,32], as well as because one is having sex with someone who is not a stranger [54], could increase the risk of transmission of BBVs/STIs. Specifically, condomless sex decision making based on the physical appearance of a partner may also be ignoring the fact that BBVs/STIs may be asymptomatic. For instance, chlamydia has been described as the most prevalent infectious disease in Australia, partly because of its asymptomatic spread among young adults [51].

Although the recent substantial decline in the prevalence of STIs in Australia has been attributed to COVID-related restrictions [55], it is likely that such declines may also be tied to a corresponding drop in the number of people presenting for screening. Evidence from the USA showed a higher positivity rate in STI screenings during the COVID-19 pandemic—despite a drop in the number of screenings—in comparison with the pre-pandemic period [3]. Thus, considering that there was a decline in condom use among sexually active young people in Australia during the pandemic [51], the possibility of an undetected spike in BBVs/STIs cannot be ruled out. This calls for increased surveillance and continuous public health education to improve awareness and empower young people with the requisite knowledge and skills to make safer sex decisions in or out of a pandemic.

## 6. Conclusions

It may be that the COVID-19 pandemic could be reinforcing pre-pandemic condomless sexual practice among participants. Considering that knowledge, though important and necessary, does not guarantee behaviour change, it is important to re-think and re-tool sex education and interventions to also consider the heuristics that shape young people’s sexual decisions and practices. This is more so in a pandemic that may be threatening the progress that has been made in controlling the epidemics of sexually transmissible BBVs and infections. As we continue to respond to COVID-19, it is critical to consider multi-faceted public health strategies that focus on controlling SARS-CoV-2 as well as other health issues such as unsafe sexual practices that could increase the spread of BBVs/STIs.

## Figures and Tables

**Table 1 ijerph-19-06264-t001:** Socio-demographic characteristics of participants.

Variable	Female (%)	Male (%)	Total (%)
** *Age* **			
18–24 years	4 (44.4%)	4 (57.1%)	8 (50.0%)
25–29 years	4 (44.4%)	2 (28.6%)	6 (37.5%)
30–39 years	1 (11.1%)	1 (14.3%)	2 (12.5%)
** *Marital Status* **			
Single	8 (88.9%)	7 (100%)	15 (93.8%)
Married	1 (11.1%)	-	1 (6.2%)
** *Background* **			
East Asia	5 (55.6%)	2 (28.6%)	7 (43.8%)
Sub-Saharan Africa	4 (44.4%)	5 (71.4%)	9 (56.2%)

## Data Availability

De-identified data are available on reasonable request.

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
