# Peer review of "“Compared to COVID, HIV Is Nothing”: Exploring How Onshore East Asian and Sub-Saharan African International Students in Sydney Navigate COVID-19 versus BBVs/STIs Risk Spectrum"

_ijerph, 2022, doi:10.3390/ijerph19106264_

Round 1
Reviewer 1 Report
Review ijerph-1673395
In the manuscript”” Compared to COVID, HIV is nothing”: exploring how onshore international students in an Australian city navigate COVID-19 versus BBV/STI risk.” Okeke reports on how the COVID-19 pandemic is affecting the protective sexual practices and potential spread of sexual transmittable diseases (STD) among international university students in Sydney, Australia. Based on a sample size of 16 the author shows an increase in casual sexual hook-ups some of which were condomless. The author argues for better public health strategies to be developed to counter potential spread of STDs.
A major flaw in the study design is the limited the sample size of only 16 participants (only 9 women). Arguably a much larger study group should have been recruitable in a university of a major city. Especially with data collection ongoing for some months. Therefore, any results are highly prone to selection bias. Omission to establish on how representative the study group is in the first place, it seems questionable to draw any conclusion, that aren’t anecdotally.
Furthermore, only recruiting students from East Asian or sub-Saharan African countries seems arbitrarily. As stated by the author, change in sexual behaviour should or should not affect the whole campus.
This lastly highlights the insufficient diligence throughout the whole manuscript. To name only a few other examples:
- Lack of adequate data representation in tables and figures. Most interview information is presented unstructured and questions of how much participants were affected by which degree of e.g., stress are answered only broadly.
- No information is given by which definition “high level of emotional distress” was measured in the first place (such as structured questioner or use of score).
- The result section is specked with interpretations.
- Any of the points stated above should have been at least discussed in the limitations section.
I therefore recommend rejecting this article for publication.
Reviewer 2 Report
I have read your manuscript““Compared to COVID, HIV is nothing”: exploring how onshore international students in an Australian City navigate COVID-19 versus BBV/STI risk” with great pleasure. This is an interesting article. It is helpful for the reader to further understand the risk of BBVs/STIs during the COVID-19 pandemic.
I have several comments:
- In “Abstract”, BBVs/STIs need to display their full name.
- In “Results”, SSA and EAS need to display their full name when it is first displayed in the manuscript.
- Please do not cite the literature in the “Results section”, please move it to the “Introduction section”.
- There are many punctuation marks that need to be rechecked.
Reviewer 3 Report
The result of this study showed how mental health distress associated with social isolation and COVID anxiety was common among the East Asian and sub-Saharan African international university students (n=16) in Sydney. Study also focused on how the participants using sex as a strategy for relief from pandemic-related mental distress and its associated BBVs/STIs risks.
- How about the pandemic related stress among the local (Australian) students?
- The sample size 16 is very low, which may not reflect the opinion of larger population.
- is there evidence or data exist to prove the inverse correlation exist between condom usage and BBVc/STIs?
Reviewer 4 Report
I have read with interest this paper. I believe that the paper is interesting. However, I have some concerns that are reported herein.
Authors should include more information about research team and reflexivity, ie: Which author/s conducted the interviews? What experience or training did the researcher have?
Related to methodological orientation and theory : What methodological orientation was stated to underpin the study? e.g. grounded theory, discourse analysis, ethnography, phenomenology, content analysis
Related to setting and sample: How were participants selected? e.g. purposive, convenience, consecutive, snowball. How were participants approached? e.g. face-to-face, telephone, mail, email. How many participants were in the study? How many people refused to participate or dropped out? Reasons? Where was the data collected? e.g. home, clinic, workplace. Was anyone else present besides the participants and researchers?
The interview guide should be included.
Related to data analysis: How many data coders coded the data? Did authors provide a description of the coding tree? Did participants provide feedback on the findings?
Round 2
Reviewer 1 Report
Please note the attached file.

Reviewer 4 Report
The manuscript has been improved and it can be published.
Author Response
Thank you so much for your time and graciousness in sharing knowledge towards the improving this manuscript. I am grateful for your review.